# Associations Between 24-Hour Movement Behaviors and Health-Related Quality of Life in East Asian Children

**DOI:** 10.3390/healthcare13192411

**Published:** 2025-09-24

**Authors:** Zhen Cui, Hongzhi Guo, Yue Wang, Jiameng Ma, Ho Jin Chung, Qiang Wang, Michael Yong Hwa Chia, Hyunshik Kim

**Affiliations:** 1Department of Physical Education, North Anhui Health Vocational College, Suzhou 234000, China; cuizhen@wbws.edu.cn; 2Graduate School of Human Sciences, Waseda University, Tokorozawa 3591192, Japan; hz-guo@moegi.waseda.jp; 3Faculty of Sports Science, Sendai University, Shibata-machi 9891693, Japan; s23510213@sendai-u.ac.jp (Y.W.); jm-ma@sendai-u.ac.jp (J.M.); 4National Institute of Education, Nanyang Technological University, Singapore 637616, Singaporemichael.chia@nie.edu.sg (M.Y.H.C.); 5College of Sports Science, Shenyang Normal University, Shenyang 110034, China; 13804999441@163.com

**Keywords:** 24 h movement behaviors, health-related quality of life, East Asian, school age children, well-being

## Abstract

**Background**: It is important to investigate the combination of 24 h movement behaviors (24 h MB)—such as physical activity, screen time, and sleep—as children’s daily habitual behaviors significantly influence their health-related quality of life (HRQoL). However, such studies remain extremely limited in East Asia. This study aimed to examine whether adherence to the 24 h MB is associated with HRQoL among children from three East Asian countries. **Methods**: In this cross-sectional study, data were collected from primary school children aged 7–12 years in Japan (*n* = 786), China (*n* = 1246), and South Korea (*n* = 1011). HRQoL was assessed using the KIDSCREEN-27 questionnaire, while physical activity, screen time, and sleep duration were measured through a self-reported questionnaire survey. **Results**: Logistic regression analyses, adjusted for relevant covariates, confirmed significant associations with HRQoL subfactors including physical well-being, psychological well-being, social support, and peer relationships. **Conclusions**: Our findings highlight the critical relationship between adherence to 24 h MB and HRQoL in East Asian children, contributing valuable evidence to support integrated health promotion strategies in this region.

## 1. Introduction

Health-related quality of life (HRQoL) is defined as a multidimensional construct that encompasses the physical, emotional, mental, social, and behavioral components of children’s lives and serves as an important indicator of the ultimate goal of good health [1]. In particular, the importance of HRQoL in terms of improving the health of populations by identifying health inequalities and planning preventive strategies accordingly has been widely recognized by researchers and policymakers, with a recent focus on assessing HRQoL in children [2,3,4]. While many studies have examined uncontrollable parental influences on children’s HRQoL—such as socioeconomic status [5], experience of illness [6], high levels of stress [7], and mental health problems [8]—a recent surge in research on the association between children’s health behaviors and HRQoL has been observed. Children with unhealthy behaviors have been reported to have lower HRQoL, which can lead to poor physical fitness and increased obesity [9]. Health-related behaviors in adulthood stem from behaviors formed in early childhood, making this a critical period to foster healthy behaviors [10,11]. Furthermore, time use—a major determinant of lifestyle—has a unidirectional or bidirectional association with health in this age group, resulting in consequences potentially persisting into adulthood. Given that time use in a day is a modifiable factor, devoting time to foster healthy behaviors to improve HRQoL is of utmost importance [12].

In 2016, the Canadian Society of Exercise Physiology (CSEP) proposed the Canadian 24-h Movement Behaviors (24-h MB) for children and adolescents aged 5–17 years [13]. These guidelines outline how children should use their time during the day to improve and maintain optimal health, such as by increasing physical activity (PA), reducing screen time (ST), and ensuring adequate sleep duration (SD) at regular times of the day. These healthy movement behaviors have been reported to be strongly associated with children’s immune systems, as well as their physical, psychological, and mental health [14,15,16]. Insufficient physical activity, excessive screen time, and inadequate sleep should all be identified as unhealthy behaviors within the framework of the 24-h MB, as each of these behaviors can independently and collectively influence children’s health-related quality of life [13,14]. However, the guidelines acknowledge that time spent engaging in one behavior affects other behaviors throughout the day, and that combinations of movement behaviors are interdependent, meaning that their collective impact over a 24 h period significantly influences children’s health [17,18]. Thus, all three types of movement behaviors (PA, ST, and SD) must be considered in relation to health over a 24 h period, as each can be repeated daily, become habitual, and even become a way of life for children. The combinations of movement behaviors within these guidelines should also be investigated to improve HRQoL among children from three East Asian countries.

Population surveillance in East Asia shows very low combined adherence to the integrated 24-h movement recommendations (≥60 min/day MVPA, ≤2 h/day screen time, and age-appropriate sleep): about 10.5% of Japanese primary school children, 7.3% of school-aged children in China, and <1% of South Korean children (7–18 years) meet all three recommendations [19,20,21]. This pattern underscores the need for population-specific evidence and targeted interventions. Unique sociocultural factors in East Asia—such as intense academic pressure, urban walkability, and high digital connectivity—are key determinants of children’s 24-h MB. These context-dependent influences underscore the need for cautious interpretation of cross-cultural comparisons [22,23,24]. However, despite variations in the way children from different cultures are assessed with regard to their health and well-being [25], studies of the association between HRQoL and 24-h MB have mostly been conducted in Western countries [26,27,28], with few studies being reported on children in East Asia. Given that foundational guideline development has occurred primarily in Western countries, generating robust evidence within the unique context of East Asia is a key rationale for the present study. Thus, further research is needed to address this important knowledge gap in relation to HRQoL. To bridge this knowledge gap, this study adopted Bronfenbrenner’s Ecological Systems Theory as its theoretical framework [29]. This theory explains that children’s behavioral development is shaped not only by individual characteristics, but also through the complex interactions between families and educational institutions (microsystem), communities (mesosystem), and national policies as well as sociocultural environments (macrosystem). The distinct macrosystems in East Asia, shaped by systematic public health policies, unique educational frameworks, and rapid socioeconomic changes, create varied contexts for children’s lifestyles. Consequently, these contextual differences may lead to cross-national variations in the relationship between adherence to the 24-h Movement Guidelines and HRQoL.

The aim of this study is to examine whether adherence to 24-h MB is associated with HRQoL in three East Asian (Japanese, Chinese, and Korean) children aged 7–12 years. These findings aim to contribute to the current evidence base on 24-h MB and HRQoL in children from three East Asian countries.

## 2. Materials and Methods

### 2.1. Study Design and Participants

For this cross-sectional study, data were collected from children participating in the Asian Children’s Lifestyle and Physical Fitness study, which recruited primary school children aged 7–12 years from representative cities (Sendai, Shenyang, and Seoul) in three East Asian countries (Japan, China, and South Korea). The inclusion criteria were as follows: children had to be aged 7–12 years, generally healthy, and without any physical or mental disabilities. In accordance with the Declaration of Helsinki, written study information and informed consent forms were provided to caregivers in each country prior to participation, and only those who agreed to participate were included in the analyses.

We also used a stratified convenience sampling strategy to recruit children from each country, considering their gender, age, and geographical location when drawing the samples. The survey was conducted using a structured questionnaire that was translated forward and backward into each country’s language. The data collection for this study included data from Japan (*n* = 1019), China (*n* = 1402), and South Korea (*n* = 1197), and the total data (*n* = 3618) were analyzed after excluding participants (*n* = 93) who did not consent to the study and erroneous data (*n* = 482), such as incomplete questionnaires (*n* = 3043). The final sample was analyzed after excluding participants with incomplete HRQoL questionnaires, missing data on key covariates, or implausible responses (*n* = 482). This study was approved by the Ethics Committee for the Use of Human and Animal Subjects for Educational and Research. The flow chart shows the progression of participants throughout the study (Figure 1).

### 2.2. Measurements

#### 2.2.1. Health-Related Quality of Life

HRQoL was measured using the non-preference-based KIDSCREEN-27 questionnaire in this study [30]. It comprises a total of 27 items and 5 sub-domains: physical well-being (5 items), psychological well-being (7 items), parent relations and autonomy (7 items), social support and peers (4 items), and school environment (4 items) [31]. Items are rated on a 5-point Likert scale, where 1 = never, 2 = seldom, 3 = quite often, 4 = very often, and 5 = always—reflecting the frequency of behaviors or feelings—or where 1 = not at all, 2 = slightly, 3 = moderately, 4 = very, and 5 = extremely—reflecting the intensity of a belief or attitude in the previous week. T-scores with a mean of 50 and a standard deviation of 10 were calculated according to the manual. The mean KIDSCREEN-27 score of 50 ± 10 defines the mean and standard deviation for children and adolescents across Europe, with higher scores indicating better levels of HRQoL [1]. The KIDSCREEN-27 questionnaire, a common HRQoL tool for children and adolescents, was developed simultaneously in 13 European countries, has been cross-culturally translated into 15 languages, and is widely used globally. With the permission of The KIDSCREEN Group [32,33], this study used questionnaires in Japanese [34], Chinese [35], and South Korean [36].

#### 2.2.2. Movement Behaviors

PA was measured using a reliable and validated item for moderate-to-vigorous physical activity (MVPA), which is a mandatory item in the internationally used Health Behaviour in School-aged Children (HBSC) study [37,38,39,40]. To help the children understand MVPA, the questionnaire defined it as “any type of PA that increases heart rate and breathing rate over a period of time, including physical education, sports, sports training, and a variety of regular daily activities such as brisk walking, hiking, and picnicking.” The question was: “How many days in the past week did you engage in MVPA for at least 60 min?” Respondents were required to select one of the following options: 0 = none, 1 = 1 day, 2 = 2 days, 3 = 3 days, 4 = 4 days, 5 = 5 days, 6 = 6 days, and 7 = 7 days. According to the 24-h MB, to meet the PA guidelines, participants must report 7 days of at least 60 min of MVPA per day [13].

We defined ST as time spent engaging in screen-based behaviors, including recreational, stationary, sedentary, and active screen time [41]. We sought to determine how much time children spent watching TV/video or using electronic devices (e.g., smartphones, tablets, computers) in their free time over the past week. The questionnaire asked two questions: (1) “On an average day in the past week, how much television or video did you watch?” and (2) “On an average day in the past week, how much did you use your smartphone, tablet, or computer?” Answers were recorded in free form, separating weekdays from weekends. Following the procedure outlined in previous research [42], average daily screen time was calculated as ([screen time during weekdays × 5] + [screen time during the weekend × 2])/7.

SD was assessed using the following questions: (1) “What time did you go to bed last week, on average?” and (2) “What time did you wake up last week, on average?” Following the procedure outlined in previous research, records were divided into weekdays and weekends using free form [42]. Total sleep duration was calculated as ([weekday sleep duration × 5] + [weekend sleep duration × 2])/7. Following previous research [13], to evaluate the adherence to the 24-h MB among study participants, the following standards for each movement behavior were applied:

(1) PA: MVPA for 60 min per day; (2) ST: less than 2 h per day; (3) SD: 9–11 h of sleep within a 24 h cycle.

#### 2.2.3. Demographic Variables

Anthropometric measurements were performed by teachers, school nurses, and researchers. Height and body mass were measured to the nearest 0.1 cm and 0.1 kg, respectively, without shoes and with participants wearing light clothing. In addition, to compare body mass between different age groups, body mass index (BMI) percentiles were categorized by sex and age, according to the 2000 Centers for Disease Control and Prevention (CDC) growth charts for the United States, into the following categories: underweight, healthy weight, overweight, and obese [43]. Fat mass (FM) was calculated according to the formula used by previous studies [44,45]. The participants’ gender, age, spectacles-wearing status, sports club membership, and number of siblings were also recorded.

### 2.3. Statistical Analysis

Data from 3043 primary school students (786 in Japan, 1246 in China, and 1011 in South Korea) who provided complete information on the study variables were analyzed using respective statistical models. While a formal a priori sample size calculation was not conducted, the large sample size (N = 3043) provided sufficient statistical power for the analyses. The adequacy of the sample for our logistic regression models was further confirmed using the events-per-variable (EPV) criterion, with all models exceeding the recommended threshold of EPV ≥ 10.

In the first model, differences in demographic variables between countries were compared using analysis of variance (ANOVA) for continuous variables such as height, body mass, BMI, and FM, and chi-squared tests for categorical variables such as gender, age, glasses-wearing status, sports club membership, and number of siblings.

In the second model, the percentage of children in each country who adhered to the 24-h MB was calculated for each movement behavior and all their combinations. Specifically, frequency analyses were used to determine the percentage of children meeting the PA, ST, and SD guidelines or combinations of these guidelines (PA + ST, PA + SD, ST + SD, and PA + ST + SD).

In the third model, independent *t*-tests were employed to compare the HRQoL subfactor scores with the fulfilment of each behavior (PA, ST, SD) of the 24-h MB. Because multiple comparisons were performed across the five HRQoL subscales, we applied a Bonferroni correction to control for Type I error. Accordingly, the threshold for statistical significance was adjusted to *p* < 0.01.

Finally, we conducted logistic regression models of the HRQoL subfactors (physical well-being, psychological well-being, parent relations and autonomy, social support and peers, and school environment) adjusting for confounders such as country, gender, age, BMI, and FM to assess the impact of 24-h MB. *p* values were set at <0.05 for statistical significance. Data analysis was performed using IBM SPSS version 27.0 (IBM, Armonk, NY, USA).

## 3. Results

Table 1 depicts the overall characteristics of participants in each country. With regard to age, Japan (17.4%) had the highest proportion of participants aged 8 years, while China (22.8%) and South Korea (22.4%) had the highest proportion of participants aged 11 years (*p* < 0.001). China had the highest proportion of tall participants (139.7 ± 12.6 cm), while South Korea had the highest average body mass among participants (38.2 ± 9.5 kg), BMI (19.5 ± 2.9 kg/m^2^), and fat mass (10.9 ± 4.4 kg) (*p* < 0.001). South Korea also had the highest proportions of participants who wore spectacles (28.8%) and were sports club members (31.5%) (*p* < 0.001).

The prevalence of children meeting each combination of 24-h MB for each country is depicted in Figure 2. Overall, 3% of children met all three recommendations, with China having the highest rate at 5.2%. In contrast, 26.8% of children did not meet any of the three recommendations, with South Korea having the highest rate at 37.5%. Considering each behavior individually, SD and PA had the highest and lowest overall fulfilment rate at 51.9% and 10.3%, respectively. With regard to countries, China had the highest rates for meeting three recommendations: ST, ST and SD, PA and SD, and PA and ST.

Table 2 depicts the difference in mean HRQoL by adherence to each recommendation. Compared with not meeting PA recommendations, meeting PA recommendations was associated with higher scores for physical well-being (62.93 ± 10.09 vs. 58.02 ± 9.68, *p* < 0.001) and social support and peers (51.28 ± 9.26 vs. 49.30 ± 9.16, *p* < 0.001). Participants who met the ST recommendations had higher physical well-being scores (60.47 ± 10.26 vs. 56.97 ± 9.38, *p* < 0.001). Participants who met the SD recommendations had higher scores for physical well-being (59.98 ± 10.07 vs. 57.95 ± 9.74, *p* < 0.005) and psychological well-being (52.84 ± 9.25 vs. 51.83 ± 8.33, *p* < 0.002). After applying the Bonferroni correction, most associations remained statistically significant. However, the association between meeting the sleep duration recommendation and the HRQoL subscale social support and peers did not retain significance (original *p* = 0.041; Bonferroni-adjusted threshold *p* < 0.01).

Figure 3A depicts the association between HRQoL and 24-h MB combinations using logistic regression analysis adjusted for country, gender, age, BMI, and FM as covariates. Among the HRQoL subfactors, physical well-being was significantly associated with all seven combinations of 24-h MB, while psychological well-being was significantly associated with all combinations of 24-h MB except PA + ST. Compared to children who met PA recommendations, those who did not had higher values for social support and peers (odds ratio [OR] 1.427, 95% confidence interval [CI]: 1.181, 1.723), with significant associations also being observed for the PA + ST combination (OR 1.536, 95% CI: 1.180, 1.999) and PA + SD combination (OR 1.410, 95% CI: 1.092, 1.819).

Figure 3B shows the results of the logistic regression analysis for the association between the number of 24-h MB guidelines met and the subfactors of HRQoL. This analysis was adjusted for country, gender, age, BMI, and FM. Overall, we observed a pattern whereby the odds of higher HRQoL increased with the number of recommendations met. Among the HRQoL subfactors, physical well-being showed a statistically significant positive association with meeting two (OR 1.816, 95% CI: 1.490, 2.199) and three of the 24-h MB (OR 4.472, 95% CI: 2.735, 7.328). Psychological well-being (OR 1.983, 95% CI: 1.169, 3.369) and social support (OR 1.512, 95% CI: 1.045, 2.179) were significantly and positively associated with meeting all three 24-h MB.

## 4. Discussion

This study examined whether compliance with the 24-h MB combination was associated with HRQoL in three East Asian (Japanese, Chinese, and South Korean) children aged 7–12 years. Our significant findings were consistent with those of previous studies, demonstrating that meeting all 24-h MB was not only associated with better HRQoL in children from three East Asian countries [27,28] but also significantly associated with the HRQoL subfactors of physical well-being, psychological well-being, and social support and peers.

Our results suggest that meeting all of the recommendations, rather than meeting one, two, or more of the 24-h MB, would be most beneficial for children’s HRQoL. Therefore, taking an integrated view of all movement behaviors throughout the day in order to improve HRQoL is essential. Furthermore, the low proportion of children meeting all of the behavioral guidelines in this study (3%) underscores the need for specific, targeted interventions for PA, ST, and SD that focus on improving the rate of meeting each of these guidelines. This requires families, schools, and communities to work together to develop programs and environments that make it easier for children to enjoy a healthy SD, limit ST during play and leisure, and engage in a range of PAs, using a social-ecological model [46,47].

Another notable finding was that physical well-being, a subfactor of HRQoL, was significantly associated with each of the seven movement behaviors outlined in the 24-h MB. Although meeting all 24-h MB resulted in the highest percentage increase in physical well-being, this is an important finding that suggests that meeting PA, SD, and ST guidelines as individual movement behaviors may also be beneficial in improving HRQoL. Previous research has also highlighted the benefits of PA, with children who maintain an active lifestyle reportedly enjoying better physical well-being than those who do not [48,49]. Moreover, the lack of adequate SD can affect the body’s immune system [50], and increased use of TV, video, and mobile devices can lead to problems with physical well-being such as weight gain [17]. These results are consistent with previous studies revealing that PA, ST, and SD are movement behaviors that are strongly associated with physical well-being throughout the day [51].

In contrast, studies on Chinese children have found that ST and SD do not significantly impact physical fitness [52], while those that have examined the association between PA, ST, and HRQoL in children have generally found consistent links between the variables. In particular, this study found an average obesity rate of 14.7% in children from three East Asian countries, but according to a report by the United Nations Children’s Fund (UNICEF), the obesity rate among school-aged children in East Asia increased from 10.4% to 27.5% in 2000, the steepest increase in the world [53]. As obesity is an important health determinant of physical well-being, the application of obesity prevention programs is also necessary to improve HRQoL [54]. However, because studies examining the association between SD and HRQoL are few and report mixed results [55], and because individual or combined behaviors are not equally important in predicting physical well-being within the constraints of a 24 h day, further research is needed to gain deeper insights into their potential impact on primary school populations.

Next, psychological well-being, another subfactor of HRQoL, was significantly associated with sleep and a combination of two or more 24-h MB. Among these, meeting SD recommendations is an important behavior that influences more than one recommendation, supporting previous research based on the 24-h MB framework linking it to psychological well-being [56,57]. Adequate sleep during childhood is crucial because children who are sleep-deprived may feel tired during their daily routines, which can affect their ability to perform physical tasks such as engaging in physical fitness [58]. This finding is also consistent with previous research revealing that meeting two or more recommendations is associated with psychological well-being. For example, more PA and better SD in a 24-h day are beneficial for psychological well-being, while higher SD is detrimental for psychological well-being [57].

However, this study reveals that the proportion of participants meeting each recommendation varies widely: 51.9% of children from three East Asian countries in this study met the SD recommendations, while 41.3% met the ST recommendations and only 5.7% met the PA recommendations. According to the 2023 World Happiness Report, Japan ranks 47th in happiness, South Korea 57th, and China 72nd [59]. To improve the psychological well-being of children from three East Asian countries, efforts should be made to not only limit play and ST during leisure time but also develop environments where children can engage in a variety of PAs, among other components of the 24-h MB. Finally, among the HRQoL subfactors, social support and peers were significantly associated with a combination of PA and two or more 24-h MB. To date, social support and peers have primarily been studied in relation to individual movement behavior, with positive relationships having been observed between PA [60] and SD [61] and negative relationships with ST [62]. However, in our study, not only PAs during movement behaviors, but also PA + ST, PA + SD, and all combinations were significantly associated, which may play an important role in improving HRQoL in children from the three East Asian countries studied.

In particular, our findings confirm the high participation rates of children in Japan and South Korea, where the participation rate in sport is around 30%. Organized sports activities are reported to be more beneficial for children than unstructured activities, and this may have influenced the results by providing more positive peer interactions [63,64]. In general, children’s health behaviors are strongly supported by their parents in the early primary school years, with peer support increasing as children progress through school due to the need to gain acceptance and respect from peers [65]. The association between peer support and sleep problems was stronger in younger than in older children [66]. Moreover, because peer victimization and bullying do exist and can affect health behaviors, it is crucial to identify and improve peer relationships through interactions with peers, teachers, and parents.

Our findings should be interpreted within the diverse sociocultural contexts of East Asia, as a monolithic view is insufficient. For instance, the varying intensity of supplementary education—such as the highly structured juku system in Japan, the intensely competitive hagwon culture in South Korea, and widespread private tutoring in China—likely creates different pressures on children’s time, potentially explaining national differences in adherence to physical activity and sleep guidelines. Differing national policies and environmental factors may also shape opportunities for physical activity, such as Japan’s promotion of community sports clubs compared to the uneven development of public play spaces in major Chinese cities [22,67]. These context-specific factors, combined with distinct cultural norms around digital media, underscore the need for cautious interpretation of cross-national data and may help explain the variance in guideline adherence observed in our study. Furthermore, other sociodemographic factors identified in this study may also influence HRQoL. For instance, the high rates of sports club participation in Japan and South Korea could foster greater social support and peer relationships, which are key domains of HRQoL [68,69]. Additionally, the number of siblings, reflecting different family structures, such as the legacy of China’s one-child policy, may shape a child’s social environment and time use, thereby impacting their overall well-being [70].

This study has several limitations. First, this study is limited by its cross-sectional design, which precludes causal inferences regarding the relationship between 24 h movement behaviors and quality of life, and by its sample of primary school children (aged 7–12), which limits the generalizability of the findings to other age groups, such as those in early childhood or adolescence. Second, although this study recruited samples from three countries in East Asia, they were from large cities in each country, such as Seoul, Sendai, and Shenyang, excluding children living in rural areas; thus, they are not representative of the population in each country. Economically disadvantaged children may have poorer physical health, which may directly affect their HRQoL [71], and further research is recommended for this segment of children based upon the health equity model [72]. Third, participants were recruited via stratified convenience sampling from urban schools in Japan, China, and South Korea. While this approach enabled large-scale, harmonized data collection, it does not yield nationally representative samples in each country; therefore, generalizability is limited primarily to similar urban school contexts. This study was based on child self-report data, which are prone to recall and social-desirability biases, age-related comprehension issues, and rounding/heaping of time estimates. These factors may lead to non-differential misclassification and attenuate observed associations. Also, fat mass was estimated using an anthropometry-based prediction equation, rather than a laboratory gold-standard method (e.g., DXA or multi-frequency BIA). We report fat mass as a proxy of adiposity and interpret findings with this limitation in mind. Fourth, the data for all three countries were collected in 2021, during the COVID-19 pandemic. The various public health measures, such as restrictions on outdoor activities and disruptions to school routines, likely influenced children’s 24 h movement behaviors and their HRQoL in unique ways. Therefore, the observed associations may reflect the specific circumstances of this period, and caution should be exercised when generalizing these findings to non-pandemic conditions. Finally, while the KIDSCREEN-27 has been validated in Japanese, Chinese, and Korean, we did not formally test for cross-cultural measurement invariance across our three national samples [73]. Consequently, we must assume that the HRQoL constructs are conceptually equivalent, although subtle cultural differences in item interpretation may exist. Future research should use methods such as multi-group confirmatory factor analysis (CFA) to formally establish the measurement invariance of the KIDSCREEN-27 before making direct cross-national comparisons in East Asia [74].

These findings have direct implications for policy and practice. An ecological, multi-level approach is likely to be most effective. At the school level, integrating brief in-class physical-activity breaks can mitigate prolonged sedentary time [75]. At the family level, culturally sensitive resources are needed to help parents manage screen time amidst high academic demands [76]. At the community level, improving the safety and accessibility of parks and recreational facilities is crucial to create environments that support healthy movement behaviors in children [77]. Collectively, these actions translate our interpretation into feasible pathways for implementation.

## 5. Conclusions

This study confirmed that meeting all 24-h MB recommendations was significantly associated with the HRQoL subfactors of physical well-being, psychological well-being, and social support and peers. These findings underscore the correlation between 24-h MB and HRQoL in children from three East Asian countries, contributing to the advancement of knowledge in the field of behavioral guidelines research. This study provides robust evidence to support future updates to movement behavior guidelines and to promote an integrated approach to child health promotion. Future longitudinal and interventional studies are warranted to establish the causal nature of these associations and to determine whether an integrated movement behavior approach is effective for improving HRQoL in East Asian children.

## Figures and Tables

**Figure 1 healthcare-13-02411-f001:**
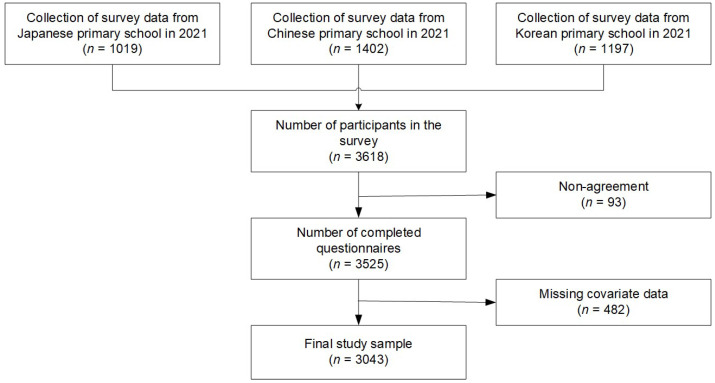
Flow chart of participants.

**Figure 2 healthcare-13-02411-f002:**
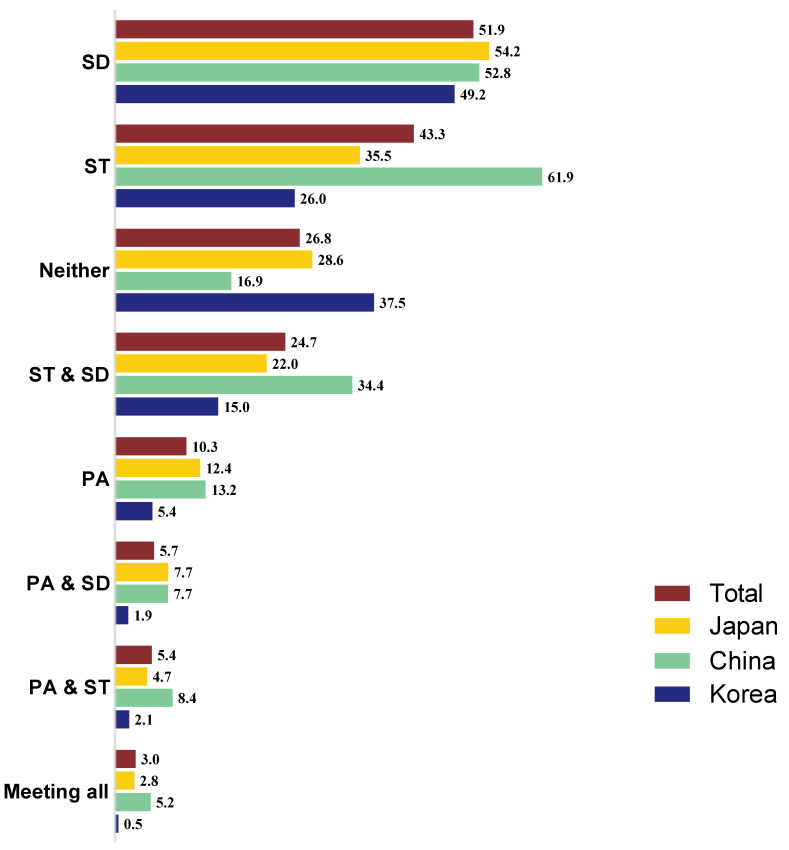
Proportion of children from three East Asian countries meeting different combinations of 24-h MB.

**Figure 3 healthcare-13-02411-f003:**
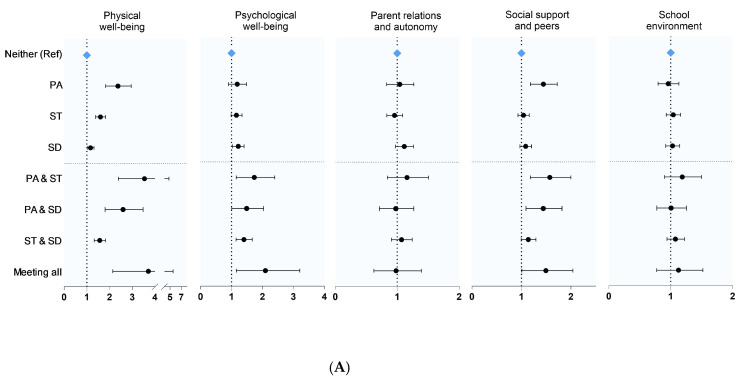
(**A**). OR and 95% CI for HRQoL according to combinations of 24 h MB met. Adjusted for country, gender, age, BMI, and FM. HRQoL was measured using the KIDSCREEN-27 questionnaire. PA: PA for 60 min per day, screen time: less than 2 h of screen time per day, sleep duration: 9–11 h of sleep within 24 h. (**B**). OR and 95% CI for HRQoL by the number of 24 h MB met. Adjusted for country, gender, age, BMI, and FM. HRQoL was measured using the KIDSCREEN-27 questionnaire. PA: PA for 60 min per day, screen time: less than 2 h of screen time per day, sleep duration: 9–11 h of sleep within 24 h.

**Table 1 healthcare-13-02411-t001:** Participant characteristics.

Variables	Japan (*n* = 786)	China (*n* = 1246)	Korea (*n* = 1011)	*p-Value*
	*n* (%) or mean ± SD ^a^			
Sex (Girls, %)	383 (48.7)	592 (47.5)	488 (48.3)	0.988
Height (cm: mean, SD)	131.4 ± 11.8	139.7 ± 12.6	139.0 ± 12.4	<0.001
Weight (kg: mean, SD)	31.2 ± 9.8	37.2 ± 12.9	38.2 ± 9.5	<0.001
BMI (kg/m^2^: mean, SD)	17.7 ± 3.1	18.5 ± 4.0	19.5 ± 2.9	<0.001
BMI z-score	0.3 ± 1.0	0.4 ± 1.2	0.9 ± 0.7	<0.001
Fat mass (kg) ^c^	8.0 ± 4.7	10.1 ± 6.5	10.9 ± 4.4	<0.001
BMI Percentile ^b^				
Under and nomal weight	574 (77.0)	1111 (64.3)	588 (58.3)	<0.001
Overweight	98 (13.2)	296 (16.6)	268 (26.6)	
Obesity	73 (9.8)	334 (19.1)	152 (15.1)	
Age (years, %)				
7	135 (17.2)	202 (16.2)	81 (8.0)	<0.001
8	137 (17.4)	266 (21.3)	177 (17.5)	
9	130 (16.5)	198 (15.9)	176 (17.4)	
10	126 (16.0)	104 (8.3)	197 (19.5)	
11	122 (15.5)	284 (22.8)	226 (22.4)	
12	136 (17.3)	192 (15.4)	154 (15.2)	
Glasses				
Yes	80 (10.3)	282 (22.6)	291 (28.8)	<0.001
No	699 (89.7)	964 (77.4)	720 (71.2)	
Sport club (Participation, %)	227 (29.0)	85 (6.8)	318 (31.5)	<0.001
Number of siblings (person)				
None	92 (12.6)	867 (70.3)	480 (47.5)	<0.001
1	346 (47.5)	349 (28.3)	472 (46.7)	
2 or more	291 (39.9)	18 (1.4)	59 (5.8)	

Note: ^a^ SD, standard deviation. ^b^ BMI percentiles of the participants were classified according to the age criteria of the growth chart by the centers for disease control and prevention in the United States [43]. ^c^ The fat mass was calculated using the height–weight equation [44]. *p*-values were calculated using *t*-test for continuous variables and chi-squared test for categorical variables.

**Table 2 healthcare-13-02411-t002:** Differences between HRQoL subfactors T-scores according to adherence to the 24-h MB.

	MVPA ^a^			Screen Time ^b^			Sleep Duration ^c^		
HRQOL Factor	Met	Not Met	*p*-Value	Met	Not Met	*p*-Value	Met	Not Met	*p*-Value
	Mean ± SD	Mean ± SD		Mean ± SD	Mean ± SD		Mean ± SD	Mean ± SD	
Physical health	62.93 ± 10.09	58.02 ± 9.68	<0.001	60.47 ± 10.26	56.97 ± 9.38	<0.001	58.98 ± 10.07	57.95 ± 9.74	0.005
Psychological well-being	52.84 ± 8.83	52.32 ± 8.87	0.328	52.71 ± 9.09	52.09 ± 8.71	0.062	52.84 ± 9.25	51.83 ± 8.33	0.002
Parent relations and autonomy	46.63 ± 7.45	46.73 ± 8.05	0.828	47.05 ± 9.04	46.57 ± 7.18	0.114	47.02 ± 8.43	46.47 ± 7.44	0.063
Social support and peers	51.28 ± 9.26	49.30 ± 9.16	<0.001	49.59 ± 9.16	49.43 ± 9.36	0.636	49.92 ± 9.54	49.22 ± 8.97	0.041
School environment	55.68 ± 10.24	56.21 ± 10.32	0.398	56.12 ± 10.14	56.04 ± 10.48	0.833	56.25 ± 10.47	55.97 ± 10.13	0.461

Note: A Bonferroni correction was applied for multiple comparisons across five HRQoL subscales. The adjusted threshold for significance was set at *p* < 0.01. SD, standard deviation. MVPA, moderate-to-vigorous physical activity. HRQoL was measured using the KIDSCREEN-27 questionnaire. ^a^ MVPA: MVPA for 60 min per day, ^b^ screen time: less than 2 h of screen time per day, ^c^ sleep duration: 9–11 h of sleep within 24 h.

## Data Availability

The data that support the findings of this study are available from the corresponding author upon reasonable request. However, the data are not publicly available due to privacy concerns and ethical considerations.

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
