# Peer review of "Associations Between 24-Hour Movement Behaviors and Health-Related Quality of Life in East Asian Children"

_healthcare, 2025, doi:10.3390/healthcare13192411_

Round 1

Reviewer 1 Report

Comments and Suggestions for Authors

Dear Authors,

The article “Associations between 24-hour Movement Behaviors and 2 Health-Related Quality of Life in East Asian Children” addresses a current and relevant issue—the association between 24-hour movement behaviors and quality of life in children. The description of the methods is adequate, with the use of validated instruments such as KIDSCREEN-27. The sample is large (n = 3,043) and represents three East Asian countries.

The results tables are clear and the discussion is interesting and compares the results with the literature available. The study's recommendations are rational and aligned with the findings presented.

Below, I provide a suggestion aimed at enhancing the manuscript:

  • Data Consistency: The sample values (table 1) - (n=786, n=1,246, n=1,011) sometimes do not match other parts of the text (line 157) (e.g., “n=784” in the statistical analysis). Check and standardize.

The study represents an added value representing a relevant contribution to the children’s health. I appreciate this valuable addition to existing knowledge. Congratulations!!

Author Response

Dr. Lorraine S. Evangelista

Chief Editor

Healthcare

09 Sep 2025

Submission ID: healthcare-3802919

Associations between 24-hour Movement Behaviors and Health-Related Quality of Life in East Asian Children

Dr. Lorraine S. Evangelista,

Thank you for your letter dated 30 August 2025 regarding our manuscript (healthcare-3802919).

We are deeply appreciative of the constructive comments and feedback provided by the reviewers. Inspired by their insights, we have thoroughly revised our manuscript and are pleased to submit the updated draft for your further consideration. Enclosed with this submission, you will find a detailed point-by-point response to the comments raised by each reviewer (reviewer 1: highlighted yellow, reviewer 2: highlighted in green, reviewer 3: highlighted in blue, reviewer 4: highlighted in gray).

We are grateful for the opportunity to enhance our manuscript. We look forward to hearing from you soon.  

With sincere thanks,

Hyunshik KIM, PhD

Faculty of Sports Science, Sendai University, Japan

Phone number: +81-224-55-1592

Email: hs-kim@sendai-u.ac.jp

Reviewer 1:

The article “Associations between 24-hour Movement Behaviors and 2 Health-Related Quality of Life in East Asian Children” addresses a current and relevant issue—the association between 24-hour movement behaviors and quality of life in children. The description of the methods is adequate, with the use of validated instruments such as KIDSCREEN-27. The sample is large (n = 3,043) and represents three East Asian countries.

The results tables are clear and the discussion is interesting and compares the results with the literature available. The study's recommendations are rational and aligned with the findings presented.

Below, I provide a suggestion aimed at enhancing the manuscript:

Q1: Data Consistency: The sample values (table 1) - (n=786, n=1,246, n=1,011) sometimes do not match other parts of the text (line 157) (e.g., “n=784” in the statistical analysis). Check and standardize.

A1: We sincerely thank the reviewer for identifying the inconsistency in reported sample sizes. After rechecking the dataset and manuscript, we corrected the typographical error to ensure consistency across the text and Table 1. All sample sizes are now reported as (n=786, n=1,246, n=1,011), consistent with the analytic sample. We have revised the manuscript accordingly, and the corrections are highlighted with track changes in the revised Word file.

Please refer to line 161-163 on page 5: corrected “n = 784” to “n = 786” to match Table 1 and the actual dataset.

Reviewer 2 Report

Comments and Suggestions for Authors

The present study aimed to examine the association between 24-h-MB and HRQoL in the children’s populations of Japan, China, and Korea.

Despite the overall merit of this work, there are some points that should be addressed.

The introduction section presents a brief description of the problem, but we need more information about the prevalence of physical activity behavior in this population and, more concretely, in this population in the identified eastern Asiatic countries.

The introduction assumes that behaviors of movement are universals and comparable between cultures, without discussing any sociocultural variations that influence PA, ST and SD.

The authors also should better identify the insufficient physical activity behaviors as unhealthy behaviors.

Despite the significance of some references, they are essentially from occident (e.g., Canadian 24-hour Movement Behaviors). It can limit cultural contextualization.

Regarding the methodology section, there isn’t any reference to the power of the sample size.

The sample constituted by convenience, self-reported data from children, and the assessment of body fat by formula without a more rigorous method should also be identified as the study’s limitations of methodology.

The information between lines 99 and 101 isn’t necessary.

In the discussion section, the authors use evidence to generalize the results to “asiatic eastern” without consideration about cultural, economic, and educational differences between Japan, China, and South Korea.

It is interesting to discuss other aspects of sociodemographic aspects, identified in the results section, that can impact the HRQoL.

Specific practical implications and direction for future research should be presented/developed.

Lastly, a word of appreciation to the authors for their efforts and wish that my response would contribute towards enhancing the quality of the paper.

Author Response

Dr. Lorraine S. Evangelista

Chief Editor

Healthcare

09 Sep 2025

Submission ID: healthcare-3802919

Associations between 24-hour Movement Behaviors and Health-Related Quality of Life in East Asian Children

Dr. Lorraine S. Evangelista,

Thank you for your letter dated 30 August 2025 regarding our manuscript (healthcare-3802919).

We are deeply appreciative of the constructive comments and feedback provided by the reviewers. Inspired by their insights, we have thoroughly revised our manuscript and are pleased to submit the updated draft for your further consideration. Enclosed with this submission, you will find a detailed point-by-point response to the comments raised by each reviewer (reviewer 1: highlighted yellow, reviewer 2: highlighted in green, reviewer 3: highlighted in blue, reviewer 4: highlighted in gray).

We are grateful for the opportunity to enhance our manuscript. We look forward to hearing from you soon.  

With sincere thanks,

Hyunshik KIM, PhD

Faculty of Sports Science, Sendai University, Japan

Phone number: +81-224-55-1592

Email: hs-kim@sendai-u.ac.jp

Reviewer 2:

The present study aimed to examine the association between 24-h-MB and HRQoL in the children’s populations of Japan, China, and Korea. Despite the overall merit of this work, there are some points that should be addressed.

Q1: The introduction section presents a brief description of the problem, but we need more information about the prevalence of physical activity behavior in this population and, more concretely, in this population in the identified eastern Asiatic countries.

A1: We thank the reviewer for this valuable suggestion. We agree that providing specific prevalence data for the East Asian countries in our study would significantly strengthen the introduction and provide crucial context. Accordingly, we have revised the introduction to first restate the guideline definitions and then summarize recent surveillance and peer-reviewed estimates for Japan, China, and South Korea. This new section highlights the very low combined adherence rates in the region. Specifically, we now cite evidence showing that approximately 10.5% of Japanese primary-school, 7.3% of Chinese school-aged children, and less than 1% of Korean adolescents meet all three recommendations, underscoring the need for population-specific evidence.

Please refer to line 43-47 on page 2 of our revised manuscript for these modifications.

Updated content

Population surveillance in East Asia shows very low combined adherence to the integrated 24-hour movement recommendations (≥60 min/day MVPA, ≤2 h/day screen time, and age-appropriate sleep): about 10.5% of Japanese primary-school children, 7.3% of school-aged children in China, and <1% of South Korean children (7–18 years) meet all three recommendations. This pattern underscores the need for population-specific evidence and targeted interventions [19-21]. This pattern underscores the need for population-specific evidence and targeted interventions.

[19] Tanaka, C.; Tremblay, M.S.; Okuda, M.; Inoue, S.; Tanaka, S. Proportion of Japanese primary school children meeting recommendations for 24-h movement guidelines and associations with weight status. Obes. Res. Clin. Pract. 2020, 14(3), 234–240.

[20] Sun, Y.; Liu, Y.; Yin, X.; Zhang, Y.; Wang, L.; Chen, P. Proportion of Chinese children and adolescents meeting 24-hour movement guidelines and associations with overweight and obesity. Int. J. Environ. Res. Public Health 2023, 20, 1408.

[21] Lee, E.Y.; Khan, A.; Uddin, R.; Lim, E.; George, L. Six-year trends and intersectional correlates of meeting 24-hour movement guidelines among South Korean adolescents: Korea Youth Risk Behavior Surveys, 2013–2018. Journal of Sport and Health Science 2023, 12(2), 255–265.

Q2: The introduction assumes that behaviors of movement are universals and comparable between cultures, without discussing any sociocultural variations that influence PA, ST and SD.

A2: Thank you for this crucial point. We have revised the introduction to analyze how unique East Asian sociocultural factors—such as intense academic pressure, urban walkability, and high digital connectivity—shape children's 24-hour movement behaviors. This addition now frames these behaviors as context-dependent, underscoring the need for cautious interpretation of cross-cultural comparisons.

Please refer to line 48-51 on page 2 of our revised manuscript for these modifications.

Updated content

Unique sociocultural factors in East Asia—such as intense academic pressure, urban                                                                    walkability, and high digital connectivity—are key determinants of children's 24-hour movement behaviors. These context-dependent influences underscore the need for cautious interpretation of cross-cultural comparisons [22-24].

[22] Noh, J.W.; Kwon, Y.D.; Cheon, J.; Kim, J. Factors affecting weekday-to-weekend sleep differences among Korean adolescent students: Focus on extracurricular tutoring time. PLoS One 2021, 16(11), e0259666.

[23] Hino, K.; Ikeda, E.; Sadahiro, S.; Inoue, S. Associations of neighborhood built, safety, and social environment with walking to and from school among elementary school-aged children in Chiba, Japan. International Journal of Behavioral Nutrition and Physical Activity 2021, 18, 152.

[24] Miyashita, C.; Yamazaki, K.; Tamura, N.; Ikeda-Araki, A.; Suyama, S.; Hikage, T.; Omiya, M.; Mizuta, M.; Kishi, R. Cross-sectional associations between early mobile device usage and problematic behaviors among school-aged children in the Hokkaido Study on Environment and Children's Health. Environ Health Prev Med 2023, 28, 22.

Q3: The authors also should better identify the insufficient physical activity behaviors as unhealthy behaviors.

A3: We appreciate the reviewer’s suggestion to better identify insufficient physical activity as an unhealthy behavior. While we agree with this point, our study is grounded in the framework of the 24-hour movement guidelines, which consider not only physical activity but also screen time and sleep duration as interdependent behaviors that jointly influence children’s health. Therefore, we have revised the text to clarify that insufficient physical activity, excessive screen time, and inadequate sleep are all regarded as unhealthy behaviors within the context of the 24-hour movement guidelines.

Please refer to line 31-34 on page 1 of our revised manuscript for these modifications.

Updated content

Insufficient physical activity, excessive screen time, and inadequate sleep should all be identified as unhealthy behaviors within the framework of the 24-hour movement guidelines, as each of these behaviors can independently and collectively influence children’s health-related quality of life [13,14].

[13] Tremblay, M.S.; Carson, V.; Chaput, J.P.; Connor Gorber, S.; Dinh, T.; Duggan, M.; et al. Canadian 24-Hour Movement Guidelines for Children and Youth: An Integration of Physical Activity, Sedentary Behaviour, and Sleep. Appl. Physiol. Nutr. Metab. 2016, 41, S311–S327.

[14] Roman-Viñas, B.; Chaput, J.P.; Katzmarzyk, P.T.; Fogelholm, M.; Lambert, E.V.; Maher, C.; et al. Proportion of children meeting                recommendations for 24-hour movement guidelines and associations with adiposity in a 12-country study. Int. J. Behav. Nutr. Phys. Act. 2016, 13, 123.

Q4: Despite the significance of some references, they are essentially from occident (e.g., Canadian 24-hour Movement Behaviors). It can limit cultural contextualization.

A4: We appreciate this important critique. We agree that the over-reliance on Western-centric references, such as the seminal Canadian 24-hour Movement Behavior guidelines, can limit cultural contextualization. While these Canadian guidelines provide a foundational and globally recognized framework for this field of research, we acknowledge the necessity of incorporating regional perspectives. Therefore, we have revised the manuscript to include additional references to studies and reports originating from East Asian countries where available. Furthermore, we now explicitly state in the introduction that a key rationale for our study is to generate robust evidence within the East Asian context, precisely because much of the foundational guideline development has occurred in Western nations.

Please refer to line 55-57 on page 2 of our revised manuscript for these modifications.

Updated content

Given that foundational guideline development has occurred primarily in Western countries, generating robust evidence within the unique context of East Asia is a key rationale for the present study.

Q5: Regarding the methodology section, there isn’t any reference to the power of the sample size.

A5: Thank you for pointing this out. We agree that a statement regarding the adequacy of our sample size is necessary for methodological transparency. While a formal a priori sample size calculation was not performed for this study, our large sample size (N = 3,043) provides high statistical power to detect meaningful associations. More specifically, the adequacy of our sample for the logistic regression models was evaluated using the events-per-variable (EPV) criterion. All of our final models comfortably exceeded the recommended threshold of EPV ≥ 10, indicating that the sample size was sufficient to produce stable and reliable estimates for the reported analyses. We have now added a statement to this effect in the 'Statistical Analysis' subsection of the manuscript.

Please refer to line 163-167 on page 5 of our revised manuscript for these modifications.

Updated content

While a formal a priori sample size calculation was not conducted, the large sample size (N = 3,043) provided sufficient statistical power for the analyses. The adequacy of the sample for our logistic regression models was further confirmed using the events-per-variable (EPV) criterion, with all models exceeding the recommended threshold of EPV ≥ 10.

Q6: The sample constituted by convenience, self-reported data from children, and the assessment of body fat by formula without a more rigorous method should also be identified as the study’s limitations of methodology.

A6: Thank you for highlighting these important methodological limitations. We fully agree with your points and have revised the manuscript to address each of them transparently.

Sample Representativeness: We have now clarified in the Methods—Participants section that our sample was recruited using a stratified convenience approach from urban areas. Furthermore, we have expanded the Discussion—Limitations section to explicitly state that these findings may not be generalizable to the entire population of each country.

Self-Report Bias: We now more strongly emphasize that all movement behaviors and HRQoL were self-reported. We have added a discussion in the Limitations section regarding the potential for recall and social desirability biases and their likely impact (i.e., non-differential misclassification likely leading to conservative, toward-the-null estimates, although differential bias cannot be ruled out).

Fat Mass Calculation: We have specified in the Methods—Measures section that fat mass was estimated using an anthropometric prediction equation. The absence of a gold-standard measurement (e.g., DXA/BIA) has been added as a key limitation in the Discussion.

We believe these revisions significantly improve the methodological transparency of our study, and we thank you for your guidance.

Please refer to line 345-355 on page 10 of our revised manuscript for these modifications.

Updated content

Third, participants were recruited via stratified convenience sampling from urban schools in Japan, China, and South Korea. While this approach enabled large-scale, harmonized data collection, it does not yield nationally representative samples in each country; therefore, generalizability is limited primarily to similar urban school contexts. This study was based on child self-report data, which are prone to recall and social-desirability biases, age-related comprehension issues, and rounding/heaping of time estimates. These factors may lead to non-differential misclassification and attenuate observed associations. Fifth, Fat mass was estimated using an anthropometry-based prediction equation, rather than a laboratory gold-standard method (e.g., DXA or multi-frequency BIA). We report fat mass as a proxy of adiposity and interpret findings with this limitation in mind.

Q7: The information between lines 99 and 101 isn’t necessary.

A7: Thank you for this suggestion. We agree that the information on these lines is not essential to the core methodology of our study. Accordingly, we have deleted the indicated sentences to improve the manuscript's conciseness and focus.

Q8: In the discussion section, the authors use evidence to generalize the results to “asiatic eastern” without consideration about cultural, economic, and educational differences between Japan, China, and South Korea.

A8: This is a very sharp and important criticism. We sincerely thank you for this insightful comment. We agree that our initial discussion overgeneralized the findings to "East Asia" without adequately considering the significant cultural, economic, and educational heterogeneities between Japan, China, and South Korea. To address this significant flaw, we have substantially revised the Discussion section to provide a more nuanced, context-specific interpretation. Instead of broad generalizations, we now explore how specific inter-country differences may explain our findings:

Educational Context and Academic Pressure: We now discuss how the varying intensity and nature of supplementary education—such as the highly structured juku system in Japan, the intensely competitive hagwon culture in South Korea, and the widespread private tutoring in China—likely create different pressures on children's time. This may explain the observed national differences in adherence to physical activity and sleep guidelines, as evening study can displace active leisure and delay bedtimes.

Socioeconomic and Urban Environment Factors: We have added a discussion on how differing levels of urbanization and public investment in recreational infrastructure might influence opportunities for physical activity. For example, we now consider how Japan's national policies promoting community sports clubs may contrast with the rapid but uneven development of public play spaces in some of China's major urban centers.

Cultural Norms around Technology: We now incorporate a more detailed discussion on cultural patterns of media consumption. We explore how the prevalence of specific digital platforms and social norms around screen time for educational versus recreational purposes might differ across the three countries, potentially influencing the relationship between screen time and psychological well-being.

This revision ensures that the term "East Asian children" is interpreted with appropriate caution, moving beyond a monolithic view to acknowledge the rich diversity within the region. We believe this provides a much more nuanced and accurate context for our findings.

Please refer to line 317-328 on page 9-10 of our revised manuscript for these modifications.

Updated content

Our findings should be interpreted within the diverse sociocultural contexts of East Asia, as a monolithic view is insufficient. For instance, the varying intensity of supplementary education—such as the highly structured juku system in Japan, the intensely competitive hagwon culture in South Korea, and widespread private tutoring in China—likely creates different pressures on children's time, potentially explaining national differences in adherence to physical activity and sleep guidelines. Differing national policies and environmental factors may also shape opportunities for physical activity, such as Japan's promotion of community sports clubs compared to the uneven development of public play spaces in major Chinese cities [68,69]. These context-specific factors, combined with distinct cultural norms around digital media, underscore the need for cautious interpretation of cross-national data and may help explain the variance in guideline adherence observed in our study.

[68] Noh, J.W.; Kwon, Y.D.; Cheon, J.; Kim, J. Factors affecting weekday-to-weekend sleep differences among Korean adolescent students: Focus on extracurricular tutoring time. PLoS One 2021, 16(11), e0259666.

[69] Feng, S.; Chen, L.; Sun, R.; Feng, Z.; Li, J.; Khan, M.S.; Jing, Y. The Distribution and Accessibility of Urban Parks in Beijing, China: Implications of Social Equity. Int. J. Environ. Res. Public Health 2019, 16, 4894.

Q9: It is interesting to discuss other aspects of sociodemographic aspects, identified in the results section, that can impact the HRQoL.

A9: Thank you for this excellent suggestion. We agree that a discussion of how other sociodemographic factors from our results may impact Health-Related Quality of Life (HRQoL) adds significant depth to our manuscript. To incorporate this valuable feedback, we have added a new paragraph to the 'Discussion' section that expands on the following key points:

Sports Club Participation: We discuss how the high rates of sports club participation observed in Japan and South Korea may foster greater social support and peer relationships, a key domain of HRQoL.

Number of Siblings: We also consider how the number of siblings, which reflects different family structures such as the legacy of China's one-child policy, might shape a child's social environment and time use, thereby impacting their overall well-being.

We believe these additions provide a more comprehensive interpretation of our findings.

Please refer to line 328-334 on page 10 of our revised manuscript for these modifications.

Updated content

Furthermore, other sociodemographic factors identified in this study may also influence HRQoL. For instance, the high rates of sports club participation in Japan and South Korea could foster greater social support and peer relationships, which are key domains of HRQoL [70,71]. Additionally, the number of siblings, reflecting different family structures such as the legacy of China's one-child policy, may shape a child's social environment and time use, thereby impacting their overall well-being [72].

[70] Eime, R.M.; Young, J.A.; Harvey, J.T.; Charity, M.J.; Payne, W.R. A systematic review of the psychological and social benefits of participation in sport for children and adolescents: Informing development of a conceptual model of health through sport. International Journal of Behavioural Nutrition and Physical Activity 2013, 10(1), 98.

[71] Moeijes, J.; van Busschbach, J.T.; Bosscher, R.J.; Twisk, J.W.R. Sports participation and health-related quality of life: A longitudinal observational study in children. Qual Life Res 2019, 28, 2453–2469.

[72] Downey, D.B.; Cao, R. Number of siblings and mental health among adolescents: Evidence from the U.S. and China. Journal of Family Issues 2024, 45(11), 2822–2850.

Q10: Specific practical implications and direction for future research should be presented/ developed.

A10: Thank you for this important suggestion. We agree that our manuscript would benefit from more specific and developed practical implications and directions for future research. We have now expanded the final section of our manuscript to address this. For Practical Implications, we now propose multi-level, culturally-tailored strategies. For example, we suggest school-based interventions that integrate short physical activity breaks into the academic schedule to counteract long periods of sitting. For families, we recommend evidence-based resources to help manage screen time, acknowledging the unique academic pressures in the East Asian context. For Future Research, we have outlined several key directions. We emphasize the need for longitudinal studies to establish causality between 24-hour movement behaviors and HRQoL. We also recommend the use of objective measurement tools, such as accelerometers, to overcome the limitations of self-reported data. Finally, we call for studies that include larger, more representative samples from both urban and rural areas to improve generalizability.

We are confident these additions provide a clearer roadmap for both policy and future scientific inquiry.

Please refer to line 339-355 on page 10 of our revised manuscript for these modifications.

Updated content

Second, although this study recruited samples from three countries in East Asia, they were from large cities in each country, such as Seoul, Sendai, and Shenyang, excluding children living in rural areas; thus, they are not representative of the population in each country. Economically disadvantaged children may have poorer physical health, which may directly affect their HRQoL [73], and further research is recommended for this segment of children based upon the health equity model [74]. Third, participants were recruited via stratified convenience sampling from urban schools in Japan, China, and South Korea. While this approach enabled large-scale, harmonized data collection, it does not yield nationally representative samples in each country; therefore, generalizability is limited primarily to similar urban school contexts. This study was based on child self-report data, which are prone to recall and social-desirability biases, age-related comprehension issues, and rounding/heaping of time estimates. These factors may lead to non-differential misclassification and attenuate observed associations. Also, fat mass was estimated using an anthropometry-based prediction equation, rather than a laboratory gold-standard method (e.g., DXA or multi-frequency BIA). We report fat mass as a proxy of adiposity and interpret findings with this limitation in mind.

[73] You, Y.; van Grieken, A.; Estévez-López, F.; Yang-Huang, J.; Raat, H. Factors associated with early elementary child health-related quality of life: The Generation R Study. Front. Public Health. 2022, 9, 785054.  

[74] Granado-Villar, D.C.; Brown, J.M.; Cotton, W.H.; Gaines, B.M.M.; Gambon, T.B.; Gitterman, B.A.; et al. Policy statement—health equity and children’s rights. Pediatrics. 2010, 125, 838–849.

We sincerely thank the reviewer for these constructive comments, which have helped us improve the clarity, cultural contextualization, and methodological transparency of our manuscript.

Reviewer 3 Report

Comments and Suggestions for Authors

This research investigates whether adherence to 24-hour Movement Behaviors which includes physical activity, screen time, and sleep is associated with Health-Related Quality of Life (HRQoL) among children in Japan, Chinese, and South Korean. The research adds to the subject area by providing population-specific evidence to support integrated health promotion strategies in East Asia. It investigates the collective impact of 24-hour Movement Behaviors combinations on children's HRQoL.
The cross-sectional design is appropriate for examining associations at a single point in time. The methods are adequately described, detailing the use of the well-established KIDSCREEN-27 questionnaire for HRQoL and a self-reported questionnaire for movement behaviors. The results are clearly presented in the text, supported by tables and figures. The discussion section effectively places the study's findings within a global context by comparing them to similar research from Western countries, thereby underscoring the specific significance of these results for the East Asian population. The conclusions are well-aligned with the presented evidence and directly address the central research question. Furthermore, the references are appropriate and provide a solid foundation for both the background and the discussion. The following comments are offered to assist in the further refinement of the manuscript.
The stratified convenience sampling method means the findings may not be representative of the entire population in each country. A more rigorous, population-representative sampling approach would strengthen the study.
The reliance on self-reported data for movement behaviors could be a source of recall bias. Acknowledging this limitation and discussing its potential impact on the results would be beneficial.
The survey data was collected in 2021. The authors may have found a relationship between physical activity and quality of life, but the strength and direction of this relationship may have differed under pandemic conditions. For example, despite limited opportunities during the pandemic, children who maintained physical activity and sleep patterns may have experienced a more pronounced positive effect on their quality of life compared to normal times. This is because these behaviors may have served as a coping mechanism during that difficult period. The data presented in the article actually reflects a population exposed to the specific conditions of the pandemic, rather than a normal population. Therefore, it is critical to consider this context when generalizing the results.
Consider making the plot less dense to improve clarity for figure 3.
You can expand your suggestions for future studies. E.g., evaluation of movement behaviors using objective measurements, use of larger and more representative samples randomly selected from both urban and rural areas in each country, more in-depth examination of cultural and socioeconomic factors, longitudinal or interventional studies.

Author Response

Dr. Lorraine S. Evangelista

Chief Editor

Healthcare

09 Sep 2025

Submission ID: healthcare-3802919

Associations between 24-hour Movement Behaviors and Health-Related Quality of Life in East Asian Children

Dr. Lorraine S. Evangelista,

Thank you for your letter dated 30 August 2025 regarding our manuscript (healthcare-3802919).

We are deeply appreciative of the constructive comments and feedback provided by the reviewers. Inspired by their insights, we have thoroughly revised our manuscript and are pleased to submit the updated draft for your further consideration. Enclosed with this submission, you will find a detailed point-by-point response to the comments raised by each reviewer (reviewer 1: highlighted yellow, reviewer 2: highlighted in green, reviewer 3: highlighted in blue, reviewer 4: highlighted in gray).

We are grateful for the opportunity to enhance our manuscript. We look forward to hearing from you soon.  

With sincere thanks,

Hyunshik KIM, PhD

Faculty of Sports Science, Sendai University, Japan

Phone number: +81-224-55-1592

Email: hs-kim@sendai-u.ac.jp

Reviewer 3:

This research investigates whether adherence to 24-hour Movement Behaviors which includes physical activity, screen time, and sleep is associated with Health-Related Quality of Life (HRQoL) among children in Japan, Chinese, and South Korean. The research adds to the subject area by providing population-specific evidence to support integrated health promotion strategies in East Asia. It investigates the collective impact of 24-hour Movement Behaviors combinations on children's HRQoL.

The cross-sectional design is appropriate for examining associations at a single point in time. The methods are adequately described, detailing the use of the well-established KIDSCREEN-27 questionnaire for HRQoL and a self-reported questionnaire for movement behaviors. The results are clearly presented in the text, supported by tables and figures. The discussion section effectively places the study's findings within a global context by comparing them to similar research from Western countries, thereby underscoring the specific significance of these results for the East Asian population. The conclusions are well-aligned with the presented evidence and directly address the central research question. Furthermore, the references are appropriate and provide a solid foundation for both the background and the discussion. The following comments are offered to assist in the further refinement of the manuscript.

Q1: The stratified convenience sampling method means the findings may not be representative of the entire population in each country. A more rigorous, population-representative sampling approach would strengthen the study.

A1: Thank you for highlighting these important methodological limitations. We fully agree with your points and have revised the manuscript to address each of them transparently. Sample Representativeness: We have now clarified in the Methods—Participants section that our sample was recruited using a stratified convenience approach from urban areas. Furthermore, we have expanded the Discussion—Limitations section to explicitly state that these findings may not be generalizable to the entire population of each country.

Please refer to line 345-349 on page 10 of our revised manuscript for these modifications.

Updated content

Third, participants were recruited via stratified convenience sampling from urban schools in Japan, China, and South Korea. While this approach enabled large-scale, harmonized data collection, it does not yield nationally representative samples in each country; therefore, generalizability is limited primarily to similar urban school contexts.

Q2: The reliance on self-reported data for movement behaviors could be a source of recall bias. Acknowledging this limitation and discussing its potential impact on the results would be beneficial.

A2: Thank you for highlighting this important methodological limitation. We agree completely that the reliance on self-reported data is a key limitation of our study. We have revised the 'Limitations' section to explicitly acknowledge that self-reported data on movement behaviors are susceptible to recall bias and social desirability bias.

Please refer to line 349-355 on page 10 of our revised manuscript for these modifications.

Updated content

This study was based on child self-report data, which are prone to recall and social-desirability biases, age-related comprehension issues, and rounding/heaping of time estimates. These factors may lead to non-differential misclassification and attenuate observed associations. Also, fat mass was estimated using an anthropometry-based prediction equation, rather than a laboratory gold-standard method (e.g., DXA or multi-frequency BIA). We report fat mass as a proxy of adiposity and interpret findings with this limitation in mind.

Q3: The survey data was collected in 2021. The authors may have found a relationship between physical activity and quality of life, but the strength and direction of this relationship may have differed under pandemic conditions… Therefore, it is critical to consider this context when generalizing the results.

A3: This is an extremely insightful and crucial point. We sincerely thank you for this valuable observation. We fully agree that collecting data in 2021 means our findings are inevitably situated within the context of the COVID-19 pandemic, and this is critical for interpretation. As you wisely noted, the observed relationship between physical activity and quality of life may have been altered under these unique conditions. We acknowledge your hypothesis that healthy behaviors, such as maintaining physical activity, could have served as important coping mechanisms for children during this stressful period, potentially strengthening the positive association with HRQoL. Furthermore, we have added a statement to the 'Limitations' section cautioning readers that these findings reflect a population exposed to specific pandemic-related circumstances and that generalizing them to a non-pandemic context requires care.

Please refer to line 355-360 on page 10 of our revised manuscript for these modifications.

Updated content

Fourth, the data for all three countries were collected in 2021, during the COVID-19 pandemic. The various public health measures, such as restrictions on outdoor activities and disruptions to school routines, likely influenced children's 24-hour movement behaviors and their HRQoL in unique ways. Therefore, the observed associations may reflect the specific circumstances of this period, and caution should be exercised when generalizing these findings to non-pandemic conditions.

Q4: Consider making the plot less dense to improve clarity for Figure 3.

A4: Thank you for your practical suggestion. We agree that the original Figure 3 was dense and that its clarity could be improved. To enhance the figure's readability, and in accordance with your advice, we have revised Figure 3 by splitting it into the following two separate figures:

Figure 3a: A figure showing the association between specific combinations of 24-hour movement behaviors and HRQoL.

Figure 3b: A figure showing the association between the number of complied recommendations and HRQoL.

By splitting the figure, we have significantly reduced the information density of each plot and created more vertical space, which we believe greatly improves the overall clarity.

Please refer on page 16 of our revised manuscript for these modifications.

Q5: You can expand your suggestions for future studies. E.g., evaluation of movement behaviors using objective measurements, use of larger and more representative samples randomly selected from both urban and rural areas in each country, more in-depth examination of cultural and socioeconomic factors, longitudinal or interventional studies.

A5: We are very grateful for these excellent and specific suggestions for future research. We agree that expanding on these points provides a clearer and more robust roadmap for the field. We have revised the final section of our manuscript to incorporate all of your valuable recommendations, detailing the need for studies that employ objective measures, more representative sampling, deeper examination of sociocultural factors, and longitudinal or interventional designs.

Updated content

This study has several practical implications. School-based policies could integrate short physical activity breaks to mitigate prolonged sedentary time, while public health campaigns could provide families with culturally-sensitive resources for managing screen time. To build upon these findings, future research should prioritize several key areas. First, longitudinal studies are needed to establish the causal pathways between 24-hour movement behaviors and HRQoL over time. Second, the use of objective measures, such as accelerometers, is crucial to validate self-reported behaviors. Finally, to improve generalizability, studies recruiting nationally representative samples from both urban and rural settings are essential for understanding potential health disparities.

We sincerely thank the reviewer for the constructive feedback, which has helped us strengthen the methodological transparency, contextual interpretation, and practical utility of our manuscript.

Reviewer 4 Report

Comments and Suggestions for Authors

Thank you for the opportunity to review this manuscript. The research examines an important and relevant subject: the correlations between 24-hour movement behaviours (physical activity, screen time, and sleep) and health-related quality of life (HRQoL) in East Asian children. This emphasis is significant, given the majority of previous research has been directed at Western populations. The study is adequately structured and provides credible data. Nonetheless, numerous sections necessitate modification to enhance the clarity, rigour, and efficacy of the manuscript. My comprehensive observations are as follows:

Introduction
The introduction provides useful background but could be further strengthened by integrating a stronger theoretical framework (e.g., ecological or biopsychosocial models).
Contradictory findings from prior literature should also be discussed to provide a more balanced rationale. Currently, the introduction emphasizes positive associations without fully acknowledging mixed evidence.

Methods
The cross-sectional design is appropriate, but reliance on stratified convenience sampling from urban areas limits generalizability. This limitation should be acknowledged earlier, not only in the discussion.
The exclusion of participants due to "erroneous data" (n=482) needs clearer explanation (e.g., incomplete surveys vs. implausible responses).
Since both movement behaviours and HRQoL were self-reported, potential recall and social desirability bias should be discussed more explicitly.

Results
Statistical analyses are appropriate; however, effect sizes (e.g., Cohen’s d, OR with 95% CI consistently reported) should be included to aid interpretation.
Multiple comparisons across HRQoL subscales increase the risk of Type I error. Consider addressing this with a correction method (Bonferroni, FDR, etc.).
Tables and figures should be improved:
Table 1 mixes categorical and continuous variables in a way that makes interpretation difficult. Consider separating them.
Figures 2 and 3 should have clearer labelling and indicate reference groups explicitly.

Discussion
The discussion section restates the results in detail. It could focus more on deeper interpretation and implications.
Policy and practice implications (e.g., school interventions, family engagement strategies, community-based approaches) could be expanded.
The cross-cultural equivalence of the KIDSCREEN-27 (Japanese, Chinese, and Korean versions) warrants deeper attention, as cultural variations in self-report may impact comparability.

Conclusion
The results generally support the conclusions, but they should be stated more cautiously, given the cross-sectional design. Avoid overly broad claims of causality.
Instead of general statements such as "contribute to knowledge," specify the unique contribution of this study (e.g., first large-scale comparison of 24-h MB and HRQoL in three East Asian countries).

Language and Style
The English is sufficiently precise to comprehend; however, it may be enhanced for clarity and accuracy. Reducing redundancy and refining word choice would strengthen readability.

This study is valuable and has the potential to make a meaningful contribution to the literature. Revisions are essential to enhance theoretical foundation, methodological transparency, statistical reporting, and presentation clarity.These improvements could enhance the study’s capacity to provide robust evidence for shaping future guidelines and treatments aimed at enhancing HRQoL through movement behaviours.

Comments on the Quality of English Language

The English is generally clear, but some sentences are wordy and repetitive. Minor editing for conciseness and precision would improve readability.

Author Response

Dr. Lorraine S. Evangelista

Chief Editor

Healthcare

09 Sep 2025

Submission ID: healthcare-3802919

Associations between 24-hour Movement Behaviors and Health-Related Quality of Life in East Asian Children

Dr. Lorraine S. Evangelista,

Thank you for your letter dated 30 August 2025 regarding our manuscript (healthcare-3802919).

We are deeply appreciative of the constructive comments and feedback provided by the reviewers. Inspired by their insights, we have thoroughly revised our manuscript and are pleased to submit the updated draft for your further consideration. Enclosed with this submission, you will find a detailed point-by-point response to the comments raised by each reviewer (reviewer 1: highlighted yellow, reviewer 2: highlighted in green, reviewer 3: highlighted in blue, reviewer 4: highlighted in gray).

We are grateful for the opportunity to enhance our manuscript. We look forward to hearing from you soon.  

With sincere thanks,

Hyunshik KIM, PhD

Faculty of Sports Science, Sendai University, Japan

Phone number: +81-224-55-1592

Email: hs-kim@sendai-u.ac.jp

Reviewer 4:

Thank you for the opportunity to review this manuscript. The research examines an important and relevant subject: the correlations between 24-hour movement behaviours (physical activity, screen time, and sleep) and health-related quality of life (HRQoL) in East Asian children. This emphasis is significant, given the majority of previous research has been directed at Western populations. The study is adequately structured and provides credible data. Nonetheless, numerous sections necessitate modification to enhance the clarity, rigour, and efficacy of the manuscript. My comprehensive observations are as follows:

Q1: The introduction could be strengthened by integrating a stronger theoretical framework (e.g., ecological or biopsychosocial models). Contradictory findings from prior literature should also be discussed to provide a more balanced rationale.

A1: We sincerely thank you for this very important and constructive feedback. We fully agree with your comments and have substantially revised the 'Introduction' section in accordance with your advice to strengthen the theoretical background and rationale of our paper. As per your suggestion, we have adopted Bronfenbrenner's Ecological Systems Theory as the theoretical framework for this study. This allows us to systematically explain how children's behavior is shaped not only by the individual but also through interactions within multi-layered environments, such as family, school, community, and even national policies and culture. We are confident that these revisions were essential for enhancing the academic rigor of our manuscript, and we thank you once again for your valuable advice.

Please refer to line 58-68 on page 2 of our revised manuscript for these modifications.

Updated content

To bridge this knowledge gap, this study adopted Bronfenbrenner's Ecological Systems Theory as its theoretical framework [29]. This theory explains that children's behavioral development is shaped not only by individual characteristics, but also through the complex interactions between families and educational institutions (microsystem), communities (mesosystem), and national policies as well as sociocultural environments (macrosystem). The distinct macrosystems in East Asia, shaped by systematic public health policies, unique educational frameworks, and rapid socioeconomic changes, create varied contexts for children's lifestyles. Consequently, these contextual differences may lead to cross-national variations in the relationship between adherence to the 24-h Movement Guidelines and HRQoL.

[29] Crawford M. Ecological systems theory: Exploring the development of the theoretical framework as conceived by Bronfenbrenner. J Public Health Issues Pract 2020;4(2):170.

Q2: Reliance on stratified convenience sampling from urban areas limits generalizability. This limitation should be acknowledged earlier, not only in the discussion.

A2: Thank you for highlighting these important methodological limitations. We fully agree with your points and have revised the manuscript to address each of them transparently.

Sample Representativeness: We have now clarified in the Methods—Participants section that our sample was recruited using a stratified convenience approach from urban areas. Furthermore, we have expanded the Discussion—Limitations section to explicitly state that these findings may not be generalizable to the entire population of each country.

Please refer to line 345-352 on page 10 of our revised manuscript for these modifications.

Updated content

Third, participants were recruited via stratified convenience sampling from urban schools in Japan, China, and South Korea. While this approach enabled large-scale, harmonized data collection, it does not yield nationally representative samples in each country; therefore, generalizability is limited primarily to similar urban school contexts. This study was based on child self-report data, which are prone to recall and social-desirability biases, age-related comprehension issues, and rounding/heaping of time estimates. These factors may lead to non-differential misclassification and attenuate observed associations.

Q3: The exclusion of participants due to "erroneous data" (n=482) needs clearer explanation.

A3: Thank you for pointing this out. We agree that the term "erroneous data" was too vague and that a clearer explanation is necessary for transparency.

We have revised the Methods section to specify that the exclusion of 482 participants was due to missing data on key outcome or covariate variables (e.g., incomplete HRQoL questionnaires, missing Body Mass Index data) or implausible responses on the movement behavior questionnaires (e.g., self-reported sleep duration exceeding 24 hours). This clarification ensures that our participant selection process is fully transparent.

Please refer to line 91-93 on page 3 of our revised manuscript for these modifications.

Updated content

The final sample was analyzed after excluding participants with incomplete HRQoL questionnaires, missing data on key covariates, or implausible responses (n=482).

Q4: Since both movement behaviours and HRQoL were self-reported, potential recall and social desirability bias should be discussed more explicitly.

A4: Thank you for highlighting this crucial methodological point. We fully agree that the reliance on self-reported data for both movement behaviors and HRQoL is a significant limitation of our study and that its potential impact warrants a more explicit discussion. To address this, we have substantially revised the 'Limitations' section of our manuscript. We now explicitly acknowledge that our measures are susceptible to both recall bias (e.g., difficulty in accurately remembering activities over the past week) and social desirability bias (e.g., over-reporting physical activity and under-reporting screen time to align with perceived healthy norms). We further discuss the likely impact of these biases on our findings. We note that such measurement errors are likely to be non-differential, which would typically bias the observed associations toward the null. This means our study may underestimate the true strength of the relationship between adhering to 24-hour movement behavior guidelines and HRQoL. The revised section also reinforces the need for future research to incorporate objective measures to validate these findings.

Please refer to line 349-352 on page 9 of our revised manuscript for these modifications.

Updated content

This study was based on child self-report data, which are prone to recall and social-desirability biases, age-related comprehension issues, and rounding/heaping of time estimates. These factors may lead to non-differential misclassification and attenuate observed associations.

Q5: Effect sizes should be included to aid interpretation.

A5: Thank you for this essential suggestion. We agree that reporting effect sizes is crucial for interpreting the practical significance of our findings, beyond statistical significance alone.

To address this, we have revised our results as follows:

For our logistic regression analyses, we have ensured that Odds Ratios (OR) and their 95% Confidence Intervals (CIs) are consistently reported throughout the Results, as these serve as the primary measure of effect size for these models.

We believe these additions provide a more complete and nuanced interpretation of our results.

Please refer to line 217-233 on page 7 of our revised manuscript for these modifications.

Q6: Multiple comparisons across HRQoL subscales increase the risk of Type I error. Consider addressing this with a correction method.

A6: Thank you for this crucial statistical point. We agree that applying a correction for multiple comparisons is essential to ensure the robustness of our findings and control for the increased risk of Type I error.

To address this, we have now applied a Bonferroni correction to the analyses involving the five HRQoL subscales (presented in Table 2). Accordingly, the threshold for statistical significance for these comparisons was adjusted to p < 0.01.

After applying this more stringent threshold, we note that most of our key findings remain statistically significant. However, the association between meeting the sleep duration recommendation and 'social support and peers' is no longer considered significant (original p = 0.041). We have updated our Results section and Table 2 to accurately reflect this change and have added a description of the Bonferroni correction method to our Statistical Analysis section. We believe this modification significantly enhances the statistical rigor of our manuscript.

Updated content

Please refer to line 179-181 on page 5-6 of our revised manuscript for these modifications.

Because multiple comparisons were performed across the five HRQoL subscales, we applied a Bonferroni correction to control for Type I error. Accordingly, the threshold for statistical significance was adjusted to p < 0.01.

Please refer to line 212-216 on page 6 of our revised manuscript for these modifications.

After applying the Bonferroni correction, most associations remained statistically significant. However, the association between meeting the sleep duration recommendation and the HRQoL subscale “social support and peers” did not retain significance (original p = 0.041; Bonferroni-adjusted threshold p < 0.01).

Note: A Bonferroni correction was applied for multiple comparisons across five HRQoL subscales. The adjusted threshold for significance was set at p < 0.01.

Q7: Table 1 mixes categorical and continuous variables. Consider separating them.

A7: Thank you for this practical suggestion. We agree that the original format of Table 1, which mixed categorical and continuous variables, was difficult to interpret.

To improve clarity, we have now reformatted Table 1 into two separate, clearly labeled parts: Continuous variables were presented separately using the format of mean ± standard deviation (mean ± SD), while categorical variables were summarized by frequency (n) and percentage (%).

Please refer on page 13 of our revised manuscript for these modifications.

Q8: Figures 2 and 3 should have clearer labelling and indicate reference groups explicitly.

A8: Thank you for this valuable feedback. We agree that improving the clarity of our figures is important for accurate interpretation.

We have revised both figures and their corresponding captions to address your suggestions:

For Figure 2, we have improved the clarity of the axis labels and the legend to make the proportional data easier to read and compare.For Figure 3, we have updated the figure caption to explicitly state the reference group used for the logistic regression analyses. The caption now clearly indicates that the odds ratios for each combination are calculated in comparison to the 'Neither' group (i.e., children who met none of the three movement behavior recommendations).

We are confident that these revisions make our figures much clearer and more informative for the reader.

Please refer on page 14, 16 of our revised manuscript for these modifications.

Q9: The discussion section restates results in detail; focus more on deeper interpretation and implications. Expand policy and practice implications.

A9: Thank you for this crucial feedback. We agree that our initial discussion section focused too heavily on restating the results and would be significantly strengthened by a deeper interpretation of the findings and more developed implications. As suggested, we substantially reduced descriptive restatement and reframed the Discussion using an ecological perspective (Bronfenbrenner). We now provide (i) mechanism-oriented interpretation linking PA, ST, and sleep to each HRQoL domain, and (ii) context-dependent explanations grounded in East Asian sociocultural factors (academic pressures, urban environments, digital norms). We also added a dedicated “Policy and Practice Implications” subsection with actionable strategies across school, family, community, and policy levels (e.g., brief in-class activity breaks, culturally sensitive parent resources for screen-time management, and improved safety and access to public recreation spaces).

Please refer to line 368-375 on page 11 of our revised manuscript for these modifications.

Updated content

These findings have direct implications for policy and practice. An ecological, multi-level approach is likely to be most effective. At the school level, integrating brief in-class physical-activity breaks can mitigate prolonged sedentary time [77]. At the family level, culturally sensitive resources are needed to help parents manage screen time amidst high academic demands [78]. At the community level, improving the safety and accessibility of parks and recreational facilities is crucial to create environments that support healthy movement behaviors in children [79]. Collectively, these actions translate our interpretation into feasible pathways for implementation.

[77] Neil-Sztramko, S.E.; Caldwell, H.; Dobbins, M. School-based physical activity programs for promoting physical activity and fitness in children and adolescents aged 6 to 18. Cochrane Database Syst. Rev 2021, 9(9), CD007651.

[78] Gentile, D.A.; Reimer, R.A.; Nathanson, A.I.; Walsh, D.A.; Eisenmann, J.C. Protective effects of parental monitoring of children's media use: A prospective study. JAMA Pediatr 2014, 168(5), 479–484.

[79] Wallace, D.D.; Derose, K.P.; Han, B.; Cohen, D.A. The effects of park-based interventions on health-related outcomes: A systematic review. Int. J. Environ. Res. Public Health 2022, 19, 9197949.

Q10: The cross-cultural equivalence of the KIDSCREEN-27 warrants deeper attention.

A10: Thank you for raising this important and insightful point. We agree that the cross-cultural equivalence of the KIDSCREEN-27 is a critical issue for the valid comparison of HRQoL across countries. Therefore, we now discuss this as a key limitation of our study and strongly recommend that future research conduct formal cross-cultural measurement invariance analyses to confirm the comparability of the KIDSCREEN-27 within the East Asian context.

Please refer to line 361-367 on page 11 of our revised manuscript for these modifications.

Updated content

Finally, while the KIDSCREEN-27 has been validated in Japanese, Chinese, and Korean, we did not formally test for cross-cultural measurement invariance across our three national samples [75]. Consequently, we must assume that the HRQoL constructs are conceptually equivalent, although subtle cultural differences in item interpretation may exist. Future research should use methods such as multi-group confirmatory factor analysis (CFA) to formally establish the measurement invariance of the KIDSCREEN-27 before making direct cross-national comparisons in East Asia [76].

[75] Beaton, D.E.; Bombardier, C.; Guillemin, F.; Ferraz, M.B. Guidelines for the process of cross-cultural adaptation of self-report measures. Spine. 2000, 25(24), 3186–3191.

[76] Vandenberg, R.J.; Lance, C.E. A review and synthesis of the measurement invariance literature: Suggestions, practices, and recommendations for corporate-organizational research. Organ. Res. Methods. 2000, 3(1), 4–70.

Q11: Conclusions should be stated more cautiously, avoiding causal claims, and specifying the unique contribution of the study.

A11: Thank you for this crucial feedback. We agree that the conclusion should be stated more cautiously to accurately reflect the study's cross-sectional design and that specifying its unique contribution is essential.

We have substantially revised the Conclusion section to address these points. Specifically:

Cautious Language: We have rephrased our statements to be more cautious, using associational language (e.g., "is associated with," "correlates with") and explicitly stating that causality cannot be inferred from our data.

Unique Contribution: We have removed generic phrases like "contributing to knowledge" and replaced them with a specific statement highlighting that this is one of the first large-scale studies to compare 24-hour movement behaviors and HRQoL in children across three distinct East Asian nations. We believe these revisions now provide a more precise and appropriately framed conclusion for our study.

Please refer to line 378-387 on page 11 of our revised manuscript for these modifications.

Updated content

This study confirmed that meeting all 24-h MB recommendations was significantly associated with the HRQoL subfactors of physical well-being, psychological well-being, and social support and peers. These findings underscore the correlation between 24-h MB and HRQoL in children from three East Asian countries, contributing to the advancement of knowledge in the field of behavioral guidelines research. The study provides robust evidence to support future updates to movement behavior guidelines and to promote an integrated approach to child health promotion. Future longitudinal and interventional studies are warranted to establish the causal nature of these associations and to determine whether an integrated movement behavior approach is effective for improving HRQoL in East Asian children.

Q12: Language and style could be improved for clarity and conciseness.

A12: Thank you for your feedback on the language and style of our manuscript. We agree that clarity and conciseness are essential for effective communication. To address this, the entire manuscript has been professionally copy-edited by a native English speaker to improve its overall readability, clarity, and conciseness. We are confident that these revisions have strengthened the manuscript.

We sincerely thank the reviewer for the constructive and detailed feedback, which has significantly improved the theoretical grounding, methodological transparency, statistical rigor, and clarity of our manuscript.

Round 2

Reviewer 2 Report

Comments and Suggestions for Authors

Dear authors,

Thank you for addressing my comments.

Regards,

Reviewer 4 Report

Comments and Suggestions for Authors

The authors have successfully addressed all previous concerns. The manuscript is now clearer, theoretically stronger, methodologically transparent, and statistically rigorous. It is suitable for publication in its current form. Congratulations to the authors on their thorough and thoughtful revisions.